# OFFENSIVENESS AS AN OPINION: DISSECTING POPULATION-LEVEL LABEL DISTRIBUTIONS

**Tharindu Cyril Weerasooriya[1*], Sarah Luger[2] & Christopher M Homan[1]**
[1]Rochester Institute of Technology, USA, [2] Orange Silicon Valley, USA
[*]cyriltcw@gmail.com

## ABSTRACT

**Warning:** *This paper contains language that may be offensive.*

Human annotation is an essential component for building human-in-the-loop machine learning systems (MLs). The diverse human disagreement that arises during annotation is often obscured because of majority voting label aggregation used for training MLs. When the minority opinion is removed in this process it may also extricate the sentiments held by people in minority demographics. This information is essential when MLs are used for offensive or hate speech identification as some content is offensive to only a minority. Collecting human annotations is an expensive task and it is even more challenging when collecting for minority voices. Population-level learning (PLL) utilizes unsupervised learning methods to represent populations of annotators using existing annotations. We test the viability and transparency of PLL with a large dataset of toxic content. We explore the clusters qualitatively by studying the language of the data items assigned to different clusters. In addition, we quantitatively analyze the nature of human disagreement via the data points assigned to the clusters.

## 1 INTRODUCTION - TOXICITY AS A PERSPECTIVE

A "winner-take-all" approach such as majority voting label aggregation, is often used to select each top label and can potentially hide the diversity of opinions produced by minority annotators (Ovesdotter Alm, 2011; Sabou et al., 2014; Waseem & Hovy, 2016; Plank et al., 2014; Kralj Novak et al., 2022; Wan et al., 2023) when models are trained. In annotation tasks like identifying offensive content, ties in opinions or a majority opinion that goes against the true nature of the content can be potentially dangerous, especially when the annotators are not representative of the population that the content targets (Sap et al., 2019). In this paper, we stress test PLL (Liu et al., 2019a; Weerasooriya et al., 2020; 2023) on a publicly available dataset on toxic content collected from various social media sources. We utilize the $D_{TR}$ dataset collected by Kumar et al. (2021) (more details in Appendix A.2) for this study.

## 2 METHODOLOGY - DISSECTING DISAGREEMENT

To understand the causes of disagreements between human annotators, we perform several measurements[1]. In this study, we audit the performance of the KMeans-based population-level clustering model (Weerasooriya et al., 2023). The KMeans-based model (PLL-KM) is able to semantically group data items that share the same label distribution. In our framework, we attempt to understand annotator disagreement through the perspective annotators' opinions and we evaluate our ability to understand and predict the population-level disagreement (Weerasooriya et al., 2020). We study the level of annotator agreement within the dataset using the following methods: (1) Entropy is utilized to understand the level of human disagreement with the majority label for a data item. (2) The annotator agreement against a deployed offensive language classification model, Perspective API (PAPI)[2]. and (3) Empirical analysis into the predictions from PLL-KM model.

---

[1]Experimental code available through `https://github.com/Homan-Lab/pldl_iclr_2023`
[2]`https://www.perspectiveapi.com`

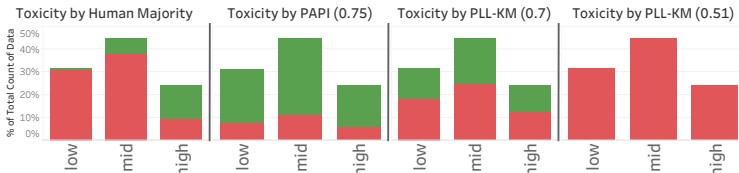

Figure 1: Cross analysis of the human toxicity classification compared with ML models. The dataset is split into three entropy levels: "low" or the majority agreeing on a single label is 0 to 0.35, "mid" is 0.35 to 0.70, and "high" or disagreement on a majority label is 0.70 to 1.05. Here **red** denotes toxic and **green** is non-toxic for a data item. A PAPI score $> 0.75$ is considered toxic.

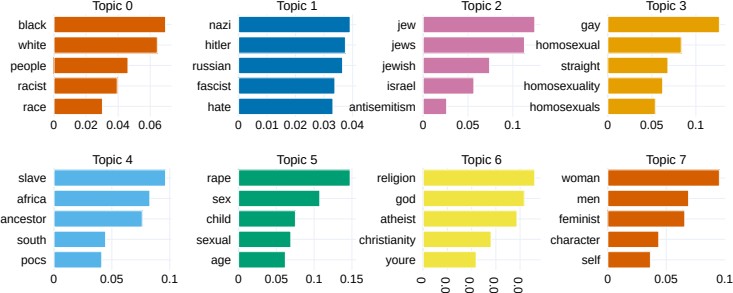

Figure 2: Analysis of word distribution for cluster #1 extracted from PLL-KM (Weerasooriya et al., 2023) predictions. Here the Y-axis contains the significant topics in each cluster and the X-axis has the corresponding c-TF-IDF score calculated by BERTopic (Grootendorst, 2022).

## 2.1 QUANTITATIVE ANALYSIS - ENTROPY

We use entropy to understand the type of the disagreement between the annotators for each data item. Since entropy is a measure of randomness in a distribution, here we use it to study how annotator opinions are scattered from the majority label. Lower entropy shows agreement of the population of annotators with the majority label and higher entropy denotes the dissonance. We explore this in Figure 1, where we bin the entropy into three categories and further study the disagreement. Overall in the dataset, most of the data points fall into the mid-level bin where there is some disagreement in the dataset. In Figure 1, the difference between the toxic classification of humans and PAPI outlines how unreliable the classification is for identification of the toxic content. The PAPI misclassified a significant portion of the content as non-toxic where it is identified as toxic by both humans and PLL-KM.

## 2.2 QUALITATIVE ANALYSIS - LANGUAGE REPRESENTATIONS

We also analyze the language of the clusters (populations of annotator pools) generated by PLL-KM using Grootendorst (2022). Figure 2 shows the most significant topics present in cluster #1 (out of the three clusters extracted with PLL-KM). As PLL methods only cluster based on the label distributions, the language topics extracted can also assist in improving the annotator pools. The topics are sorted by c-TF-IDF score for each word in the topic. The scores can be used to indicate the distinctness of each word in the cluster.

## 3 CONCLUSION AND FUTURE WORK

In this study, we utilize entropy as a metric to understand the disagreement of the human annotators against different baselines of modeling the annotator disagreements. Figure 1 shows how PLL-KM performs against the baselines of human majority and PAPI. And in Figure 2, we explore qualitatively the nature of content that PLL-KM is able to capture for clustering annotations. We explore clusters qualitatively and quantitatively; this framework can also be used for uncovering the significant information obscured through "winner-take-all" label aggregation methods. Our future work aims to understand the reasons why human majority, PAPI, and PLL disagree in judgment.

URM STATEMENT

The authors acknowledge that at least one key author of this work meets the URM criteria of ICLR 2023 Tiny Papers Track.

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

# A APPENDIX

## A.1 RELATED METHODS

Along with other methods in label distribution learning Dawid & Skene (1979); Geng et al. (2014); Liu et al. (2019c); Davani et al. (2021); Gordon et al. (2022), population-level label distribution learning (PLL) (Weerasooriya et al., 2023; Liu et al., 2019a; Weerasooriya et al., 2020; 2022) advocates for label distributions for both training and final predictions in the ML pipeline. The challenge often in such models is not having enough annotator-level data to train a model in the wild (Prabhakaran et al., 2021).

## A.2 EXPERIMENTAL DATASET - TOXIC RATINGS ($D_{TR}$) DATASET

Kumar et al. (2021) collected the dataset containing 107,620 items that are annotated by 17,280 participants. There are at least five annotators per data item. In the dataset, the content sources are Twitter (67%), Reddit (15%), and 4chan (18%) comments. The dataset contains the scores from the Perspective API[3] (PAPI) and granular annotator demographic information. The authors used the toxicity score of $> 0.75$ to indicate a comment as toxic. We utilize the same categorization in this study. The authors use five levels of toxicity in the study, (1) extremely toxic, (2) very toxic, (3) moderately toxic, (4) slightly toxic, and (5) not at all toxic.

For the analysis in our Methodology 2, we condense the five label choices as:

---

[3] https://www.perspectiveapi.com

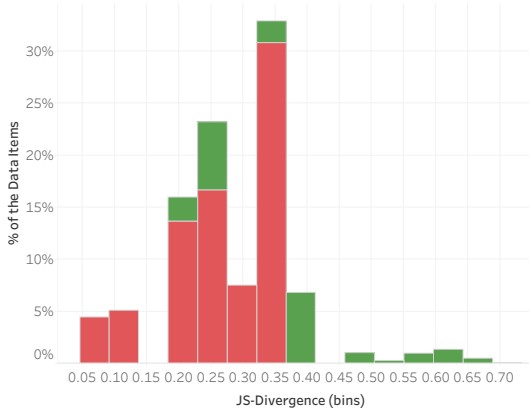

Figure 3: Histogram of the distribution of the closest 1000 items to the PLLKM predicted cluster centroid. The colors denote the human classification for the data item, where **red** denotes toxic and **green** denotes non-toxic.

- toxic - (1) extremely toxic and (2) very toxic.
- moderate or non toxic - (3) moderately toxic, (4) slightly toxic, and (5) not at all toxic.

## A.3 JS DIVERGENCE

We analyze the JS-divergence (Menéndez et al., 1997) of the dataset against the empirical result and predicted result for understanding how label distributions changed during the prediction. Liu et al. (2019b); Weerasooriya et al. (2020) utilized KL-divergence in their analysis. However, since KL is not a symmetrical measure, we utilize JS. The JS analysis is utilized as a way to understand how the PLL-KM models are able to predict the human labels.

