# OpenReview forum: "Offensiveness as an Opinion: Dissecting population-level Label Distributions"
_ICLR.cc/2023/TinyPapers — Submitted to Tiny Papers @ ICLR 2023_

### Official Review · Reviewer_2vwq · 2023-04-02

**Confidence:** 4

**Summary Of Contributions:**

 The paper proposes an approach called population-level learning (PLL) to address the issue of diverse human disagreement during annotation. The paper also highlights the importance of considering minority sentiments when identifying offensive or hate speech

**Rating:**

Great Start (GS): a submission which meets some of the reviewing criteria but has room for improvement

**Strengths And Weaknesses:**


Strengths:
- Novel approach
- Importance of minority opinions
- Usefulness in real-world applications
- Comprehensive analysis

Weaknesses:
- Absence of diversity among annotators: The findings of the authors may not be broadly applicable or generalizable because they do not disclose the demographics of their annotators.
- The proposed method is promising for representing populations with a range of perspectives, but it is uncertain how well it would function in real-world applications where there are numerous confounding variables that can impair model performance.
- It is difficult to determine whether PLL is better than existing methods because the research does not compare its results to those obtained using other unsupervised learning methods or supervised approaches for recognizing offensive or hate speech.

**Suggested Changes:**

- Addressing diversity in annotators: Providing information about the demographic makeup of annotators would help readers better understand how well PLL can represent diverse populations with different opinions.
- Discussing real-world applications: The authors should discuss potential use cases where PLL may be useful in identifying offensive or hate speech and how it compares with existing techniques.
- Comparing results with existing methods: Including a comparison between PLL and other unsupervised learning methods or supervised approaches for identifying offensive/hate speech would help assess whether this method is superior compared with others.
- Simplifying technical language - using simpler language throughout certain sections will make it easier for readers who are unfamiliar with machine learning terminology to follow along

---

### Official Review · Reviewer_fkyF · 2023-04-02

**Confidence:** 4

**Summary Of Contributions:**

The authors aim to characterize interannotator diasgreement by using k-means to cluster data samples with similar label distribution.

**Rating:**

Great Start (GS): a submission which meets some of the reviewing criteria but has room for improvement

**Strengths And Weaknesses:**

The problem is well-motivated. IAA is a notable issue in generating label sets.

The figures are potentially useful but unclear (See suggested changes) .

**Suggested Changes:**

Is Fig 1 only showing the samples classified as toxic by PLL-KM? Why should that be? Sec 2.1 needs to clarify the setting.

The x-axis should be labeled in Fig 2. The corresponding Sec 2.2 also needs clarification

Given limited space, I'd prioritize methodology over introduction and future work.

---

### Official Review · Reviewer_Wunq · 2023-04-02

**Confidence:** 2

**Summary Of Contributions:**

The paper explores the use of population-level learning in a toxicity dataset where a ‘winner takes all’ approach may discount minority opinion of annotators. The authors conduct both a quantitative and a qualitative analysis.

**Rating:**

Great Start (GS): a submission which meets some of the reviewing criteria but has room for improvement

**Strengths And Weaknesses:**

Strengths:

-	The paper provides good context for why ‘winner-take-all’ approaches may hide opinion diversity, and why this is an important problem.

-	The paper explores a variety of methodologies to understand disagreement.

Weaknesses:

-	The conclusion is not clear, and the results are not given proper context. Neither the quantitative analysis nor the qualitative provide enough detail or interpretation.

-	Not enough detail is provided for the methodologies to allow for reproduction.


**Suggested Changes:**

-	Provided clearer interpretation of the results and concluding remarks. The conclusion section is primarily a summary of the paper.

-	Provide better description/citations for the PLL-KM and PAPI

---

### Official Review · Reviewer_YQaR · 2023-04-03

**Confidence:** 2

**Summary Of Contributions:**

Human annotator disagreement is lost in majority voting label aggregation, which can particularly impact under-represented communities. The authors audit population-level learning using a KMeans-based model (PLL-KM) to semantically group data items with similar label distributions and perform two qualitative analyses to find that prior methods misclassified toxic content, and to conjecture that auditing word distributions extracted from PLL-KM can assist in improving the annotator pools.

**Rating:**

Great Start (GS): a submission which meets some of the reviewing criteria but has room for improvement

**Strengths And Weaknesses:**

Strengths
- The authors address the problem of human annotator disagreement and veracity, which is increasingly crucial given the role of human annotators in the steering of ML systems such as LLMs.
  - The perspective of majority voting label aggregation as a lossy process is intuitive yet seems under-appreciated by the ML community, further pointing to the contributions this paper could bring to the community.
- Figure 1 and 2 are very nicely formatted and have potential to convey a great deal of information, however, some of this is missing on the reader due to the issues mentioned below.


Weaknesses
- The paper is a little difficult to read, due to typos and some important components only appearing in the appendix without proper reference. Tighten up text body to pull in more content from the appendix. Refer more often to appendix for content that can’t be fit into paper.
- The contributions of the authors are not clear.
  - For example, it's unclear if PLL-KM is their work or prior work. If it is part of this work's contribution, more algorithm details are required.
  - As detailed below, the source of the ground truth labels (required by some of the paper's statements) is unclear.

**Suggested Changes:**

Reproducibility
- In the sentence: "The PAPI misclassified a significant portion of the content as non-toxic where it is identified as toxic by both humans and PLL-KM.", it is unclear where ground truth is coming from. And earlier statement mentions “true nature of the content”, but also without citing source for ground truth.
  - Ground truth seems very tricky given that even the human annotators are disagreeing.  Perhaps the sentence in the appendix that “The authors use five levels of toxicity in the study…” points toward an additional level of human eval done but the authors, but this should be clarified and details for reproducibility should be included.

Clarity
- Use a more specific term than “machine model” (e.g. something consistent with the plot titles) in Fig 1 caption
- “The KMeans-based model (PLL-KM)” is not cited.
- The lines: “(2) The annotator agreement against Perspective API (PAPI). An in the wild offensive language classification model.” are sentence fragments. Is the intention to combine the two together, separated with commas?
- “Perspective API (PAPI)” is only cited in the Appendix. Move into text body.

- Nits:
  - In the enumerated list: Use consistent sentence casing if using periods after each item, or replace the periods with a comma or semi colon.
  - “PLLKM” missing hyphen  in Fig 3
  - Align the direction of the labels tick marks in Fig 2
  - Incorrect quotation direction in Fig 1 caption: ”low” or majority agreeing on a single label is (0 to 0.35), ”mid” is (0.35 to 0.70), and ”high”
  - Incorrect quotation direction in Introduction and Conclusion for phrase: ”winner-take-all”
  - Remove “The” before “The Figure 1” in the conclusion.
  - Add “in” to “And Figure 2” in the conclusion.

---

### Meta-Review · Area_Chair_Bxxw · 2023-04-07

**Recommendation:** Invite to archive
**Confidence:** 4

**Metareview:**

1. Paper communicates clearly with appropriate literature but fails to provide the much needed references of PLL-KM & PAPI
2. Paper needs to provide information on the ground truth with respect to human annotators
3. Reproducible code, data is not provided
4. Does follow basic requirements and page limits. Keeping page limit in mind, author could have added the much needed information to make this the best paper.


**Summary:**

Population level learning KMeans based PLL-KM (unsupervised) to identify toxic content

**Comments And Feedback To The Authors:**

The Paper needs major revision for CCR. \
Kindly follow the suggested changes as mentioned below
1. Include links for Data & code in appendix for reproducibility criteria
2. Mention usage of Perspective API (PAPI) in the text body rather than only in the appendix.
3. Provide relevant citation under references for PAPI
4. Provide citation for Population level learning KMeans-based model(PLL-KM)
5. In the appendix include a short algorithmic description
6. Performance comparison of PLL-KM and other supervised or unsupervised methods
7. Make __Suggested changes__ as mentioned by the reviewers [YQaR](https://openreview.net/forum?id=DoOiwBcRir3&noteId=e8N6g6RZbD), [Wunq](https://openreview.net/forum?id=DoOiwBcRir3&noteId=OolrYceUhr)
, [fkyF](https://openreview.net/forum?id=DoOiwBcRir3&noteId=WGWg9G8e9Q)
& [2vwq](https://openreview.net/forum?id=DoOiwBcRir3&noteId=JIDjQ-sJ8a8)



**Reason For Not Giving A Higher Recommendation:**

1. Paper lacks information of how well PLL-KM performs better than other unsupervised method, a performance comparison is needed for this
2. Paper provides very little information and no citation under references on PLL_KM & PAPI


**Reason For Not Giving A Lower Recommendation:**

Agree with the comments & suggested changes with the reviewers \
The paper will be CCR ready & accepted post revision

---

### Decision · Program_Chairs · 2023-04-09

Invite to archive